



# The Global SMOS Level 3 daily soil moisture and brightness
# temperature maps
Ahmad Al Bitar[1,2], Arnaud Mialon[1,2], Yann H. Kerr[1,3], François Cabot[1,3], Philippe Richaume[1],
Elsa Jacquette[3], Arnaud Quesney[4], Ali Mahmoodi[1], Stéphane Tarot[5], Marie Parrens[1], Amen Al-Yaari[6],
Thierry Pellarin[7], Nemesio Rodriguez-Fernandez [1], Jean-Pierre Wigneron[6]
[1] Centre d'Etudes Spatiales de la Biosphère, Université de Toulouse, CNES/CNRS/IRD/UPS, Toulouse, France.
[2] Centre National de Recherche Scientifique, Paris, France.
[3] Centre National d'Etudes Spatiales, Paris, France.
[4] CapGemini Sud, 109 Avenue du Général Eisenhower, 31000 Toulouse, France.
[5] IFremer, BP 70, 29280 Plouzane, France.
[6] INRA, UMR1391 ISPA, Villenave d'Ornon, France.
[7] IGE, University Grenoble Alpes, CNRS/G-INP/IRD/UGA, Grenoble, France.
*Correspondence to*:  Ahmad Al Bitar (ahmad.albitar@cesbio.cnes.fr)
**Keywords:** soil moisture, SMOS, L-band, SMAP, Multi-orbit, microwave, retrieval algorithm
**Abstract**: The objective of this paper is to present the multi-orbit (MO) surface Soil Moisture (SM) and
angle binned Brightness Temperature (TB) products for the SMOS (Soil Moisture and Ocean Salinity)
mission based on the a new multi-orbit algorithm. The Level 3 algorithm at CATDS (Centre de Traitement
Aval des Données SMOS) makes use of multi-orbit (multi-revisits) retrieval to enhance the robustness and
quality of SM retrievals. The motivation of the approach is to make use of the temporal auto-correlation of
the vegetation optical depth (VOD) to enhance the retrievals when an acquisition occurs at the border of
the swath. The retrieval algorithm is implemented in a unique operational processor delivering multiple
parameters (e.g. SM and VOD) using angular signatures, dual polarization and multiple revisits. A
subsidiary angle binned TB product is provided. In this study the L3 TB V300 product is showcased and
compared to SMAP (Soil Moisture Active Passive) TB. The L3 SM V300 product is compared to the
single-orbit (SO) retrievals from Level 2 SM processor from ESA (European Space Agency) with aligned



configuration. The advantages and drawbacks of the Level 3 SM product (L3SM) product are discussed.
The comparison is done at global scale between the two datasets and at local scale with respect to *in situ*
data from AMMA-CATCH and USDA-ARS WATERSHEDS networks. The results obtained from the
global analysis show that the MO implementation enhances the number of retrievals up to 9 % over
certain areas.  The comparison with the *in situ* data shows that the increase of the number of retrievals
does not come with a decrease of quality. But rather at the expense of an increased lag of product
availability from 6 hours to 3.5 days which can be a limiting factor for forecast applications like flood
forecast but reasonable for drought monitoring and climate change studies. The SMOS L3 soil moisture
and L3 brightness temperature products are delivered using an open licence and free of charge by CATDS
(http://www.catds.fr).
**Abbreviations**
ARS            Agricultural Research Service
AMMA           Analyse Multidisciplinaire de la Mousson
AMSR-E         Advanced Microwave Scanning Radiometer - Earth Observing System
ASCAT          Advanced Scatterometer
CATDS          Centre Aval de Traitement des Données SMOS
CNES           Centre National d'Etudes Spatiales
CCI            Climate Change Initiative
CDTI           Centro para el Desarrollo Tecnológico Industrial
DPGS           Data Processing Ground Segment
EASE-Grid      Equal-Area Scalable Earth Grid
ECMWF          European Centre for Medium-Range Weather Forecasts
ECV            Essential Climate Variables
EO             Earth Observation
ESA            European Space Agency
IFREMER        Institut Français de Recherche pour l'Exploitation de la Mer
ISEA           Icosahedral Snyder Equal Area
L-MEB          L-band Microwave Emission of the Biosphere
MO             Multi Orbit
NASA           National Aeronautics and Space Administration (U.S.A.)
SM             Soil Moisture
SMAP           Soil Moisture Active and Passive





| 60 | SMOS | Soil Moisture and Ocean Salinity |
| 61 | SMUDP | Soil Moisture User Data Product |
| 62 | SO | Single Orbit |
| 63 | TOA | Top of Atmosphere |
| 64 | USDA | United States Department of Agriculture |
| 65 | VOD | Vegetation Optical Depth |





## 1. Introduction

Surface Soil Moisture (SM) is a control physical parameter for many hydrological processes like infiltration, runoff, precipitation and evaporation (Koster et al., 2004). Estimates of SM are needed for many applications concerned with monitoring droughts (Keyantash & Dracup, 2002), floods (Brocca et al., 2010, Lievens et al., 2015), weather forecast (Drusch, 2007, de Rosnay et al., 2013), climate (Jung et al. 2010), and agriculture (Guérif & Duke, 2000). It is identified among the 50 Essential Climate Variables (ECV) for the Global Climate Observing Systems (GCOS). It has been also selected for the creation of decadal time series from remote sensing in the European Space Agency (ESA) Climate Change Initiative (CCI) project (Hollmann et al., 2013).

SM can be obtained from several Earth Observation (EO) techniques ranging from visible to microwave using active (Ulaby et al., 1996) and passive (Kerr & Njoku. 1990) instruments. Retrieval of SM from microwave sensors is a challenging exercise because features like surface heterogeneity (water surfaces, land use), vegetation cover (vegetation density and distribution), climatic conditions (freezing, snow), acquisition configurations (angle, frequency, polarisation), and topography (multiple scattering) need to be carefully considered while upscaling to the sensor coarse resolution. Several approaches like regression models (Njoku et al., 2003, Wigneron et al., 2004 and Saleh et al., 2006), statistical and contextual methods (Verhoest et al., 1998), neural networks (Liu et al., 2002, Rodriguez-Fernandez et al., 2015) and radiative transfer based approaches (Kerr & Njoku, 1990, Wigneron et al., 2007, Owe et al., 2008, O'Neill et al., 2013) have been developed to retrieve SM based on the sensor frequency, acquisition modes and richness of information (multi angular, full polarization, active). The Soil Moisture and Ocean Salinity (SMOS) mission of ESA (Kerr et al., 2001, 2010) with contributions from Centre National d'Etudes Spatiales (CNES) in France and Centro para el Desarrollo Tecnológico Industrial (CDTI) in Spain is the first earth observation mission dedicated for SM mapping. The SMOS Level 2 (L2) SM retrieval algorithm (Kerr et al., 2012) uses the L-band Microwave Emission of the Biosphere (L-MEB) radiative transfer model (Wigneron et al., 2007) as a forward operator in association with the Levenberg-Marquardt optimization algorithm to retrieve physical parameters, mainly SM and VOD.



The L-MEB radiative transfer model is based on the optical depth single scattering albedo ($\tau$–$\omega$) model
(Mo et al., 1982) combined to specific parameterisations to take into account the impact of vegetation and
soil roughness on polarization mixing and angular signature.  The Soil Moisture Active Passive (SMAP)
mission, launched by NASA on January 2015 delivers TB observations on a fixed (40°) incidence angle
(Entekhabi et al. 2010). The SMAP soil moisture processor currently relies on a Single Channel Algorithm
(SCA) (O'Neill et al., 2012) for its main product. This algorithm uses a forced vegetation optical thickness
in a single-orbit configuration. Miernecki et al. (2015) presented a review and a comparison of the
different retrieval approaches for L-Band microwave from EO missions (SMOS, SMAP, AQUARIUS).
Passive microwave sensors have a high revisit frequency: 1 day for Advanced Microwave Scanning
Radiometer - Earth Observing System (AMSR-E) (Njoku & Entekhabi , 1996), and 2-3 days for SMOS
and SMAP. In this study the multiple orbit (MO) , multi-angular and dual channel (H/V) operational
retrieval algorithm implemented at the CATDS (Centre Aval de Traitement des Données SMOS) by
Centre National d'Etudes Spatiales (CNES) is presented.  Retrieval using temporal series is becoming
increasingly common in operational EO retrieval algorithms for optical and to some extent microwave
technologies. Some examples in the optical domain are the correction of aerosols impact for visible
images (Hagolle et al., 2008, 2015), the cloud detection (Hagolle et al., 2010) and the use of multiple
revisits for land cover classification (Inglada & Mercier, 2007). The previous methodologies are being
implemented for high-end level 2-A and level 3 products for the Copernicus Sentinel-2 mission. The use
of multiple revisits in the radar community is a standard approach.  The SM retrievals from ERS,
Advanced Scatterometer (ASCAT), RADARSAT-2 and Sentinel-1 are based on a change detection
algorithm (Wagner et al., 1999, 2013; Naeimi et al., 2009). Similarly, Mattia et al. (2006) introduced a
priori surface parameters and multi-temporal Synthetic Aperture Radar (SAR) data to remove the impact
of vegetation and soil roughness in SM retrieval from SAR. Recently a generalization of change detection
to multiple regression using Cumulative Distribution Function (CDF) transformations was applied to
RADARSAT-2 time series data and validated over the Berambadi watershed, South India (Tomer et al.,
2015). In microwave  radiometry, Konings et al. (2016) presented a time series retrieval of vegetation
optical depth based on AQUARIUS L-Band acquisitions.

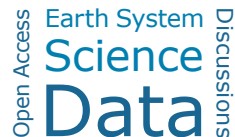

Here a detailed presentation of the products and retrieval algorithm and an inter-comparison between the
SMOS SO (Single orbit) and the SMOS MO (Multi-orbit) operational products is done.    More
specifically, the objective of this paper is to present the daily L3 SM and TB V300 products and
associated algorithms and to compare the SMOS MO level 3 retrievals to the level 2 single-orbit
operational retrievals that are were obtained using V600 L1 ESA-SMOS products.   Since the SMOS
mission launch in November 2009, this is the first reprocessing to have an aligned version of the
processors from Level 1 up to Level 3 enabling a direct comparison of the products. In the next sections,
the multi-orbit retrieval SM algorithm and the angle binned TB are presented. The datasets used for the
assessment, the results of the comparison and conclusions are presented.
**2. The CATDS Level-3 soil moisture processor**
**2.1 Algorithm overview**
The Level-3 SM (L3SM) processor is a set of several algorithms. The forward model in L3SM uses the
same physically based forward models as the ESA SMOS Level 2 SM processor, but in a MO retrieval
context. A short summary of the main features of this processor is provided hereby, a detailed description
is provided in (Kerr et al. 2012). The SMOS L2 retrieval can be divided into two main components:
1) The first component is a physical model that computes TB at the antenna reference frame forced by
ancillary data (land classification, soil properties) and physical parameters (skin or near-surface
temperature and soil temperature).. The selected physical model for the SMOS mission is L-MEB from
Wigneron et al. (2007). The main features of the L-MEB physical model implementation in the SMOS
operational processor are:
• Effective scattering albedo is considered.
• SM and VOD are jointly retrieved over nominal (bare soil and low vegetation) surfaces using
angular signature information.
• Dual polarization is used. Full polarisation data is only used to take into account the Faraday
rotation and geometric rotation to transform modelled TB from the Top Of Atmosphere (TOA) to
the antenna reference frame.



• The mean antenna pattern (Kerr et al., 2012) is used in the iterative retrieval algorithm. The mean

weighting function expresses the average contributions for all angular acquisitions. The -3 dB

footprints is about 20 km in radius. This corresponds to the nominal resolution of the synthetic

aperture. Also this corresponds to 86% of the signal if a homogeneous surface is considered (Al

Bitar et al., 2012).

• Surface heterogeneity is considered through aggregated TB contributions from $4 \times 4$ km² surface

units. The contributions are then convoluted by the mean antenna pattern. A total area of $125 \times$

125 km² is considered at each retrieval node to compute the total contributions.

• Dynamic changes in surface state (freezing, rainfall…) are considered through the use of ancillary

weather data from ECMWF (European Centre for Medium-Range Weather Forecasts) reanalysis

products.

Since the mission launch many improvements have been implemented in the operational processing
model see for instance the improved parametrization of the forest albedo in Rahmoune et al., (2014) or
the choice of dielectric mixing models in Mialon et al., (2015).
2) The second component of the retrieval algorithm is an iterative optimization scheme that minimises a
Bayesian cost function constructed from the observed and the modelled TBs in order to retrieve the
physical parameter values.  Pre-processing and post-processing steps are implemented to filter the input
and output data for undesired effects, like the decrease of quality due to spatial sampling or Radio
Frequency Interferences (RFI) (Oliva et al., 2012, Richaume et al., 2014).
The physical approach at Level-3 MO is the same as that of Level-2 SO. In fact the core processing uses
the same implementation of the L-MEB radiative transfer model. The main difference in Level-3 is the use
of several orbits, rather than one, to retrieve SM and VOD. This has an impact first on the post-processing
steps for selecting the orbits and second on the optimization scheme to retrieve the parameters. Since the
Level-2 retrieval is a multi-parameter retrieval, the Level-3 is thus a multi-orbit multi-parameter retrieval.
The reasons that motivated the use of the MO approach are the following:



• The angular sampling and radiometric accuracy at the border of the swath is reduced. Figure 1
shows the cumulative number of records (TB$_X$, TB$_Y$, TB$_{XY}$) for several descending orbits. The
asterisk in the images represents the same location in La Plata region, South-America. The reddish
region observed on 18$^{th}$, 20$^{th}$ and 23$^{rd}$ of May 2010 shows the decrease of number of TB
measurements during the instrument calibration phases. But most important is the smaller number
of TB measurements (35) on the same location observed on the 21th of May image. A low
number of TB measurements spanning a narrow range of incidence angles can make the iterative
estimation of SM and VOD to fail. The use of MO can help improving the number of successful
retrievals at the border.

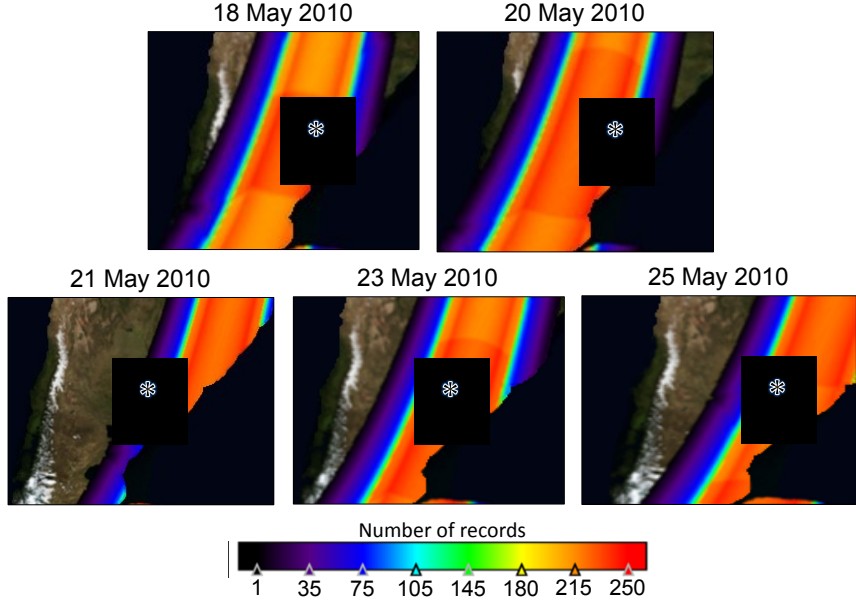


*Figure 1 - Number of TB records across the swath for a period of 8 days - from 18 May*
*2010 to 25 May 2010 - over the area of La Plata Argentina.*
• The VOD is expected to vary slowly in time and thus to be highly correlated between two
consecutive ascending or descending orbits or over short period of time (few days). In fact at L-
band the VOD is mainly correlated to vegetation water content (Jackson & Schmugge, 1991).



Other general motivations for Level-3 products are to provide a global gridded product, in contrast to
swath based products and to provide fixed angle binned TB products. The 25 km Equal-Area Scalable
Earth Grid version 2.0 (EASE-Grid 2.0) (Brodzik & Knowles, 2002) which was selected for the 3 MO
product has also a spatial sampling closer to the sensor nominal resolution.
**2.2 Orbit selection**
The selection of orbits is needed to filter TBs at high latitudes where a sub-daily revisit is available and to
generate the time series dataset on the EASE-Grid 2.0 as input to the MO retrieval. The following criteria
are applied for the selection of revisits:
• Ascending and descending orbits are processed separately, since the impact of RFI (Oliva et al.,
2012) and sun corrections (Khazaal et al., 2016) between ascending and descending orbits are
very different.
• TB products are generated from the snapshot based L1B products which are TBs in the Fourier
domain. This consists in an Inverse Fast Fourier Transform (IFFT) to make the transition from the
Fourier domain to the spatial domain using the L3 EASE-Grid 2.0. In a subsequent step, TBs
measurements corresponding to the same grid point are selected from the different snapshots (for
a given grid point, the incidence angle of the observation is different for each snapshot) to
construct a grid-point-based product similar to the ESA L1C TB product but in EASEv2 grid. The
alternative of interpolating the ESA L1C TB dataset from the 15 km Icosahedral Snyder Equal
Area (ISEA) grid to the 25km EASE-Grid 2.0 grid. This option was excluded because it can
generate interpolation artefacts on the TB products that would have propagated through the
processing chain.
• TB products are filtered at high latitudes where more than one revisit per day occurs (latitudes
above 60°N and 60°S). A maximum of one revisit per day is considered. The selection criterion is
the minimum distance from the centre of the swath because the radiometric accuracy and
resolution is best at the centre. This criterion is applied for each grid node individually.



At this level the acquisitions for a given day for ascending and descending orbits are separately stored in a
3 dimensional matrix accounting for snapshots, longitude and latitude. A snapshot is an image associated
to the acquisition of SMOS during a given integration time (epoch). Snapshots have different epochs and
polarization following a preprogramed acquisition sequence. From this product a fixed angle binned TB
product is generated as presented in Section 3. The product is also used in the next processing steps of
L3SM MO.
•    For each retrieval and over each node a 7-days period is considered in which 3 revisits are
selected when more are available. The first coincides with the central date (date of main product).
The two others correspond to selected dates either before (previous 3.5 days) or after (3.5 days
posterior) the considered date. Like in the previous processing step, the selection is done based on
minimum distance from the swath centre for each node.

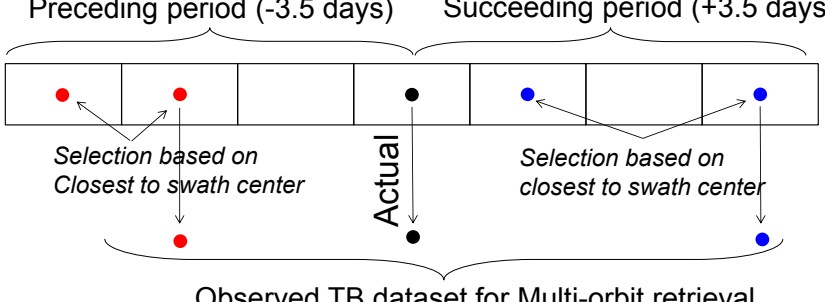


*Figure 2 - Selection of revisit orbits for the multi-orbit retrieval at SMOS CATDS.*
**2.3 Cost function and retrieval**
Observed TB at antenna reference frame from the "precedent", "actual" and "succeeding" dates are
assembled for each node. The forward algorithm is run to generate the modelled TB for each of the TB
dataset records. The ancillary data and parameters are considered for each record independently.   A
Bayesian cost function that includes the aforementioned MO data, namely observed and modelled TB, is
then constructed. This is achieved by incorporating in the retrieval approach a temporal auto-correlation
function for the VOD. The cost function is as follows:



$$Cost = (TB_M - TB_F)^t \cdot COV_{TB}^{-1} \cdot (TB_M - TB_F) + \sum_p (P - P_0)^t \cdot COV_p^{-1} \cdot (P - P_0) \qquad (1)$$

Where $COV_{TB} = \sigma_{TB}^2$ is the error covariance matrix of TB data by assuming no auto-temporal correlation,
$TB_M$ is the measured TB from SMOS, $TB_F$ is the forward modelled TB using L-MEB, $P$ is the retrieved
parameters (SM,VOD), $COV_p$ is the error covariance matrix for parameter $P$. $P_o$ is the a-priori value of
parameter $P$.
It is important to note that three SM values are retrieved simultaneously at each node: $SM_P$ for the
preceding date, $SM_A$ for the actual date and $SM_F$ for the succeeding date. The same applies to VOD. In the
case of SM, the a-priori values are given from ECMWF reanalysis data.
When P = $SM_P$, $SM_A$ or $SM_F$, the error covariance matrix considering no-cross or auto-correlation is given
by:
$$COV_{SM} = \sigma_{SM0}^2 \cdot I \qquad (2)$$

where $\sigma_{SM0}^2$ is the standard-deviation error associated to SM. It is set to a high value: 0.7 m³/m³. **I** is the
(3×3) identity matrix.
When P = VOD the error covariance matrix, considering temporal auto-correlation and no-cross
correlation between the different parameters is given by:
$$COV_{VOD} = \sigma_{VOD0}^2 \begin{bmatrix} 1 & ... & ... \\ \rho(t_P, t_A) & 1 & ... \\ \rho(t_P, t_F) & \rho(t_A, t_F) & 1 \end{bmatrix} \qquad (3)$$

Where $\sigma_{VOD}^2$ is the standard-deviation error associated to VOD, and ρ the correlation function modelled
assuming a Gaussian auto-correlation distribution:
$$\rho_{VOD}(t_1, t_2) = \rho_{max}(t_1, t_2) \cdot exp\left(-\frac{(t_1 - t_2)^2}{Tc^2}\right) \qquad (4)$$

Where $t_1$ and $t_2$ are the time (expressed in days) corresponding to the VOD retrievals dates (P, A or F),
$\rho_{max}(t_1, t_2)$ is the maximum amplitude of the correlation function between $t_1$ and $t_2$, $Tc$ is the characteristic
correlation time for VOD ($Tc$ = 30 days for forests and TC = 10 days for low vegetation).



Figure 3 shows the shape of the correlation function for the two correlation lengths used in the processing.
The green curve corresponds to the forested surfaces and the blue one to the nominal surfaces (bare soil
and low vegetation).

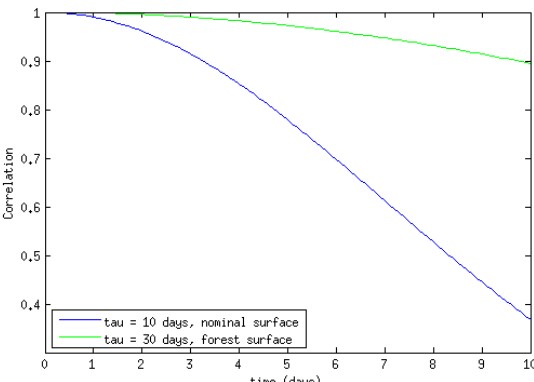

*Figure 3 Auto-correlation functions for vegetation optical depth (VOD) for different*
*correlation lengths (green: forested surfaces, blue: nominal surfaces).*
The parameter values namely ($SM_P$, $SM_A$, $SM_F$, $VOD_P$, $VOD_A$ and $VOD_F$) are retrieved by minimising the
cost function in an iterative procedure using the Levenberg-Marquardt optimisation algorithm.  So, at the
end of each daily retrieval, three SM values are available. The retrieval associated to the best goodness of
fit ($X^2$) value is then selected and delivered in the 1 day product. This product is only available when the
filtering is finished, and thus with 7 days of lag time. Using the daily maps, time synthesis products (3
days, 10 days and monthly) are then provided. A detailed description of the algorithm is presented in the
CATDS L3 Algorithm Theoretical Basis Document (Kerr et al.,2013).
**3. The CATDS Level-3 angle binned TB processor**
The objective of this algorithm is to generate a product containing fixed angle full polarization brightness
temperatures at Top of Atmosphere (TOA) but with the polarizations expressed in the ground reference
frame (horizontal and vertical components) over the EASE-Grid 2.0. The main input to this algorithm is
the dataset of snapshots mentioned in the previous section. The algorithm consists of four steps: (a)
filtering, (b) interpolation, (c) reference transformation and (d) angle binning. However note that before
being projected to a ground frame, the data is processed in the instrument reference frame. Thus TBs are



labelled $TB_Y$ and $TB_X$ to express that the polarisations are at satellite level while once processed they will
be provided in the ground reference frame and be labelled $TB_H$ and $TB_V$.

## 3.1 TB filtering

The filtering eliminates brightness temperatures that are impacted by anthropogenic effects (such as Radio
Frequency Interferences (RFI)), or spurious effects (such as sun impact). The filtering criteria, shown in
Table 1, are similar to those for L3 MO SM retrieval. All filtering criteria should be met, otherwise the
acquisition is discarded. In case a cross-polarisation is discarded, the associated X and Y acquisitions are
also removed.

*Table 1 – List of applied filtering criterion used on brightness temperature products prior to interpolation*

| Filtering criteria | Applied test | Filtering criteria | Applied test |
|---|---|---|---|
| thresholds | $50\ K < TB_X\ \&\ TB_Y < 340\ K$ <br> $-50\ K < TB_{xy} < +50\ K$ | RFI | L1A STRONG RFI (flag is off) |
| Amplitude | $50\ K < \sqrt{TB_x^2 + TB_y^2} < 500\ K$ | | L1B STRONG RFI (flag is off) |
| Standard deviation | $TB - 2 \cdot ATB < TB < TB + 2 \cdot ATB$ | | POINT SOURCE RFI (flag is off) |
| 1st Stokes | $ST1 - \overline{ST1} < 5 + 4 * ATB$ | | TAILS RFI (flag is off) |
| Spatial resolution † | $SMEF < (55 \times 55)\ km^2$ | Sun correction ‡ | SUN_POINT (flag is off) |
| | $LMA / Lmi < 1.5$ | | SUN_TAILS (flag is off) |
| | BORDER FOV (flag is off) | | |

*Where ATB is the radiometric accuracy of SMOS TB, ST1 is the first Stokes parameter, $\overline{ST1}$ is the average of ST1 over each dwell*
*line (angular signature), ST4 is the forth Stokes parameter, SMEF is the area of the half maximum contour of the mean synthetic*
*antenna pattern, LMA Length of the major axis of synthetic antenna pattern, Lmi Length of the minor axis of synthetic antenna*
*pattern.*
*† Spatial resolution: eliminates records that are impacted by aliasing (outside the alias free field of view).*
*‡ if active the flag means that the pixel is located in a zone where a Sun alias was reconstructed (after sun removal, measurement*
*may be degraded). The sun tail is considered when the pixel is located in the hexagonal alias directions centred on a sun alias.*

## 3.2 TB Interpolation

The acquisition sequence of SMOS is shown in Table 2. It shows that at each epoch an acquisition can be
co-polarised (X, Y) or combined cross (XY, YX) and co-polarised. The table shows that there is no



complete dataset at any epoch. A weighted linear interpolation is used to compute the missing acquisitions
based on adjacent ones.

*Table 2 - Acquisition sequences of SMOS in full polarization mode (capital letters are used for pure acquisition)*

| Snapshot number | 1 | 2 | 3 | 4 | 5 | 6 | 7 | 8 | 9 | 10 | 11 | 12 |
|---|---|---|---|---|---|---|---|---|---|---|---|---|
| TB (Real/Imaginary) | | X/XY | | Y/YX | | X/XY | | | | | | Y/YX |
| TB (co-polarisation) | X | X | Y | y | X | x | | Y | X | | Y | Y |


The weighting function accounts for the two following elements:
- The accuracy of acquisition: the TB acquisitions have different accuracy levels because the integration
time is longer when only co-polarisation is acquired (pure acquisition) compared to the case where
combined cross and co-polarisation are acquired.
- The time span of acquisition: The time span between two acquisitions of the same mode is not constant.
Acquisitions closer in time are considered more reliable than farther ones taking into consideration that the
synthetic antenna function is rotating and that the incidence angle is changing.
The time interpolation function of TB at time i ($TBi$) is as follows:

$$
\begin{cases}
TB_i = \dfrac{W_{i-1}.TB_{i-1} + W_{i+1}.TB_{i+1}}{W_{i-1} + W_{i+1}} \\[2mm]
W_{i-1} = \dfrac{1}{\sigma_{i-1}.nb\_epo_{i-1}} \\[2mm]
W_{i+1} = \dfrac{1}{\sigma_{i+1}.nb\_ep_{i+1}}
\end{cases}
\tag{5}
$$


Where $nb\_epo_i$ is the number of epochs between acquisitions at time $i$, $\sigma$ is the associated radiometric
accuracy, $W_i$ is the weighting coefficient at time $i$. The standard deviation of the interpolated field is
computed based on the square root of the weighted variances of the adjacent acquisition. We assume that
the acquisitions are not-correlated, therefore no cross correlation term is considered in the equation. The
following formulation is used:

$$
\begin{cases}
\sigma_i = \sqrt{\dfrac{(Q_{i-1}.\sigma_{i-1})^2 + (Q_{i+1}.\sigma_{i+1})^2}{Q_{i-1}^2 + Q_{i+1}^2}} \\[2mm]
Q_i = \dfrac{1}{nb\_epo_i}
\end{cases}
\tag{6}
$$




The same approach as eq.(5) while applying a constant weight is used to compute the interpolated values
of auxiliary information like major and minor semi-axis length, incidence angle, Faraday angle and
geometric angle.
**3.3 Transformation from antenna to ground reference frame**
In this step, the TBs are transformed from antenna reference frame (X,Y) to the ground reference frame
(H,V). This is done without accounting for atmospheric and galactic contributions. They are considered as
TOA TBs. The TB components at antenna reference frame exhibit polarisation mixing due to the geometry
of the acquisition (Figure 4). Faraday rotation will also alter slightly the polarisations.

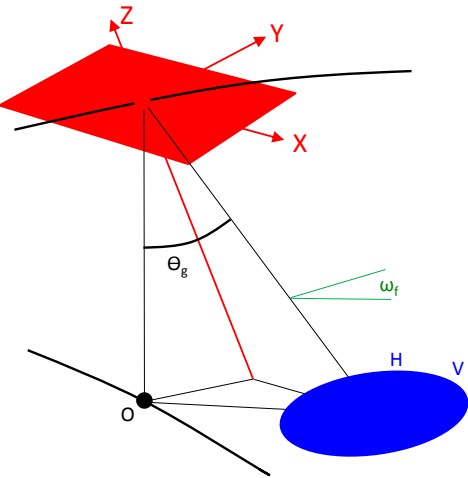


*Figure 4 - Transformation from antenna (S) to ground reference frame (G), $\omega_f$ is the faraday rotation angle and $\Theta_g$ is the geometric rotation angle (adapted from SMOS L2 ATBD).*

The inverse of the rotation matrix is used to transform the TB data from antenna to ground reference
frame:

$$
\begin{bmatrix} TB_H \\ TB_V \\ TB_3 \\ TB_4 \end{bmatrix} = IRM \begin{bmatrix} TB_X \\ TB_Y \\ 2 \cdot reel(TB_{XY}) \\ -2 \cdot imag(TB_{XY}) \end{bmatrix}
$$

(7)

$TB_3$ and $TB_4$ are the Stokes 3 and Stokes 4 components. The Inverse of Rotation Matrix (IRM) is given by:

$$
IRM = \begin{bmatrix} \cos^2 a & \sin^2 a & \cos a . \sin a & 0 \\ \sin^2 a & \cos^2 a & -\cos a . \sin a & 0 \\ -\sin 2a & \sin 2a & \cos 2a & 0 \\ 0 & 0 & 0 & 1 \end{bmatrix}
$$

(8)



Where $a = \Theta_g + \omega_f$          (9)
With $\Theta_g$ being the geometric angle and $\omega_f$ being the Faraday rotation angle as shown in Figure 4.



The accuracies of the TB data are then computed by propagating the accuracies using the above matrix:
$$\begin{cases} \sigma TBH = \left(IRM_{1,1}^2 \cdot \sigma TB_X^2 + IRM_{1,2}^2 \cdot \sigma dTB_Y^2 + 4 \cdot \left(IRM_{1,3}^2 + IRM_{1,4}^2\right) \cdot \sigma TB_{XY}^2\right)^{0.5} \\ \sigma TBV = \left(IRM_{2,1}^2 \cdot \sigma TB_X^2 + IRM_{2,2}^2 \cdot \sigma dTB_Y^2 + 4 \cdot \left(IRM_{2,3}^2 + IRM_{2,4}^2\right) \cdot \sigma TB_{XY}^2\right)^{0.5} \\ \sigma TB3 = \left(IRM_{3,1}^2 \cdot \sigma TB_X^2 + IRM_{3,2}^2 \cdot \sigma dTB_Y^2 + 4 \cdot \left(IRM_{3,3}^2 + IRM_{3,4}^2\right) \cdot \sigma TB_{XY}^2\right)^{0.5} \\ \sigma TB4 = \left(IRM_{4,1}^2 \cdot \sigma TB_X^2 + IRM_{4,2}^2 \cdot \sigma dTB_Y^2 + 4 \cdot \left(IRM_{4,3}^2 + IRM_{4,4}^2\right) \cdot \sigma TB_{XY}^2\right)^{0.5} \end{cases} \quad (10)$$

Where $IRM_{i,j}$ are the $i^{th}$ column and $j^{th}$ line components of the IRM matrix
**3.4 Angle binning**
This step consists of averaging the TOA TBs at fixed angle intervals using an arithmetic mean. The
selected incidence angle bins, shown in Table 3, are designed to cover also the SMAP acquisition angle
(40°).

*Table 3 - Selected incident angle bins*

| Bin id | 1 | 2 | 3 | 5 | 6 | 7 | 8 | 9 | 10 | 11 | 12 | 13 | 14 |
|---|---|---|---|---|---|---|---|---|---|---|---|---|---|
| Bin centre | 2.5° | 7.5° | 17.5° | 22.5° | 27.5° | 32.5° | 37.5° | 40° | 42.5° | 47.5° | 52.5° | 57.5° | 62.5° |
| Bin width | 5° | 5° | 5° | 5° | 5° | 5° | 5° | 5° | 5° | 5° | 5° | 5° | 5° |


All TB values outside the interval defined by mean (TB) ± 2 std (TB), where is the standard deviation of
TB for each angle bin (not to be confused with the radiometric accuracy), are considered as outliers and
removed from the binning. This helps the removal of the low RFI effects and other undesired impacts. If
one component of TB ($TB_H$, $TB_V$, $TB_{HV}$) is filtered out, all the other components are disregarded.



**4. Datasets**
**4.1 Remote sensing datasets**
**4.1.1 SMOS CATDS Level 3 soil moisture products**
The CATDS Level 3 user data products (CLF3UA/D) are MO soil moisture retrieval products. They
contain 1 day global maps of geophysical parameters (SM, VOD, imaginary and real dielectric constant
part…) computed as described above, processing parameters (percentage of forest cover, type of surface
model…) and quality indicators (Probability of RFI, goodness of fit $X^2$ …) over continental surfaces for
ascending and descending orbits separately. They are in the NetCDF format over the EASE-Grid 2.0 25
km. They are generated at the Institut Français de Recherche pour l'Exploitation de la Mer (IFREMER)
for CNES and distributed via the CATDS webportal (http://www.catds.fr) and ftp server. The operational
production of L3SM started in 2010 and it is currently ongoing. The time span used in this study covers
2010 - 2015 for the global maps and 2010 - 2016 for the time series analysis. The user has access to the
latest versions of the products either from reprocessing or from operational processing. The current study
uses the latest data corresponding to reprocessing RE04 which uses CATDS V300 corresponding to ESA
V620 Level 1 & 2. It is the first simultaneous Level 2 and Level 3 reprocessing campaign since the start of
the mission. Previous versions of the L3SM products where compared to soil moisture products from
AMSR-E (Al-Yaari et al., 2014 a) and ASCAT (Al-Yaari et al., 2014 b) missions, but this is the first
comparison enabling a aligned configuration of the L2SM SO and L3SM MO. It has homogenized inputs
(L1B/C) and physical parametrization. It uses the Mironov dielectric constant model (Mialon et al., 2015),
enhanced forest parametrization for albedo (Rahmoune et al., 2014), enhanced global soil texture map
consistent with the one used for the SMAP mission, and latest RFI detection techniques (Richaume et al.,
2014). It uses also the latest (V620) brightness temperature products at Level 1B. The SM maps are
extracted in the present study from the L3 product. After extraction, RFI filtering is applied with
Probability of RFI < 10 % and goodness of fit with a probability of $X^2 > 0.95$.



**4.1.2 SMOS DPGS Level 2 soil moisture product**
The ESA L2 Soil Moisture User Data Product (SMUDP), which is a SO retrieval product, is used in this
study for comparison purposes. This product is a half-orbit swath based dataset of physical variables (SM,
VOD, dielectric constant imaginary and real part…), processing parameters (percentage of forest cover,
type of surface model…) and quality indicators (Probability of RFI, $X^2$, …) over continental surfaces.
Ascending and descending orbits are processed separately in the current configuration. The SMUDP
product is delivered in the BinX format over the ISEA discrete global grid (Carr et al. 1997), with a
hexagonal partitioning of aperture 4 at a resolution of 9 km known as ISEA4H9. The grid point centres
have a fixed separation distance of around 15 km. Products are generated at the ESA SMOS Data
Processing Ground Segment (DPGS) and disseminated by ESA via Earth Online. The DPGS and CATDS
share the same reprocessing dissemination strategy, and users are provided access to the most recent
products even before the end of reprocessing campaign. Version 620 of SMUDP is used in this study, and
the time span selected is 2010-2015 for the global analysis and 2010 – 2016 in the time series analysis.
The main characteristics and differences between the L2SM SO retrieval and L3SM MO retrieval
products are summarised in Table 4.
**4.1.3 SMOS CATDS Level 3 brightness temperature products**
The SMOS CATDS full polarisation angle binned daily brightness temperature product (CDF3TA/D)
version 310 were downloaded from the same database as the L3 MO SM. These products consist of global
1 day maps of full polarisation TB over fixed angle bins with their associated accuracies. Detailed
computation was described above in Section 3. The product also contains auxiliary data like the geometric
angles, Faraday angles, length of major semi-axis and length of minor semi-axis. Quality flags are also
provided in the product. The $TB_H$ and $TB_V$ records are extracted for the 40° bin. No additional filtering is
done over these products.
**4.1.4 SMAP NSIDC L1C brightness temperature**
The SMAP mission from NASA was launched in January 2015. It operates like SMOS in L-band using a
radiometer and a radar (that was operational for about 80 days). It has a local overpass time at 6H00 am



and 6H00 pm for ascending and descending orbits respectively but the acquisitions are not necessarily
synchronous with SMOS. In this study we use the SMAP TB derived from the radiometer acquisitions.
The SMAP L3B_SM_P product is downloaded from the National Snow and Ice Data Centre (NSIDC)
website. The SMAP L3 TB is used as input for the SM retrievals and it is corrected for the water
contribution and atmospheric effects. It is provided on the EASE 2.0 grid with a 36 km resolution product.
The data is in HDF5 format. The $TB_H$ and $TB_V$ records are extracted for year 2015.

*Table 4 – Main characteristics of the SMOS Level 3 and Level 2 SM products*

| Product | L3SM | L2SM |
|---|---|---|
| Name of product | MIR_CLF3A/D | MIR_SMUDP |
| Gridding system | EASEv2 | ISEA 4H9 |
| Product sampling | 25 km at | 15 km fixed |
| Resolution | *SMOS nominal resolution of 40km* | |
| Multi-parameter retrieval | yes | Yes |
| Angular signature | yes | Yes |
| Polarization impact | yes | Yes |
| Multi-orbit | yes | No |
| Forward model | *L-MEB (tau omega)* | |
| Availability | 3.5 - 7 days | 6 hours |
| Processing centre | CATDS (CNES) | DPGS (ESA) |
| Format | NetCDF | BinX |
| Version | V300 | V620 |
| Coverage | Global grid | Swath based |


**4.2 *In situ* datasets**
In this study, the SMOS soil moisture products are evaluated against two networks with spatially
distributed soil moisture data at the footprint scale (USDA Watersheds and AMMA CATCH). The *in situ*
soil moisture data from probes installed at near surface are used. These sites provide a soil moisture
reading, representative of the first 5 cm of the top soil layer, as they are vertically installed. This may lead
to a mismatch between the sensor sampling depth and the expected representative depth 0-2 cm or 0-3 cm
of the L-Band microwave radiometers (Escorihuela et al., 2010).  The choice of the sites is done to cover



contrasting environments over two different continents to provide an overview of the SM MO processor
performances. The statistics over the sites are computed for data available within 1 hour of space-borne
acquisitions (SMOS, SMAP).
**4.2.1- AMMA dataset**
The AMMA long term observing system (AMMA-CATCH (1996) and AMMA-CATCH (2005)) includes
three mesoscale sites located in Niger, Benin, and Mali that are representative of the West-African eco-
climatic gradient (Cappelaere et al., 2009; Mougin et al., 2009).. The AMMA-CATCH soil moisture
network is a well-established network in terms of satellite product assessment (de Rosnay et al., 2009;
Pellarin et al., 2009; Louvet et al., 2015). Niger and Benin, of the three meso-scale sites, are selected for
this study. The Niger site, centred at 13.645° N–2.632° E, is mainly composed of tiger bush on the
plateaus, fallow savannah and pearl millet crop fields on the sandy slopes (Cappelaere et al., 2009). The
Benin site, located at 1.5–2.8° E; 9–10.2° N, is mainly composed of Woody savannah and tropical forest.
Most of ground-based instruments are located in the North–West part of the Ouémé catchment (9.745° N–
1.653° E). The observed annual rainfall amount was 1578 mm in 2010, 1093 mm in 2011 and 1512 mm in

2012.

**4.2.2- USDA - WATERSHEDS**
The United States Department of Agriculture (USDA) Agricultural Research Service operates a network
of densely instrument watershed across the US. Surface soil moisture (5 cm) is monitored across the
watersheds and recorded on an hourly basis since 2002. The USDA provides estimates of the average soil
moisture over an area that has approximately the size of a SMOS footprint. Two of the watersheds have
been selected for this study: Walnut Gulch (WG), Arizona, USA (Keefer et al., 2008) and Little Washita
(LW), Oklahoma, USA (Elliott et al., 1993). Soils in WG can be classified as sandy loam. The original
datasets are available from https://www.tucson.ars.ag.gov/dap/ for WG and from
http://ars.mesonet.org/webrequest/ for LW. Over LW the soil properties are more heterogeneous with a
loam, clay and sand texture. Previous studies on calibration and scaling have quantified the uncertainty of
the *in situ* measurements over the sites to be lower than 0.01 $m^3/m^3$ when compared to gravimetric



measurements. The basin scale weighted average is based on the Thiessen polygon method and has a
standard deviations between 0.05 and 0.10 $m^3/m^3$. A detailed description of the site characteristics is
provided in Jackson et al. (2010), and details on the averaging procedure are provided in Jackson et al.
(2012). This network has been used for validation of remote sensing soil moisture datasets (including
SMOS) in many studies (Sahoo et al. 2008, Jackson et al. 2012, Leroux et al. 2014). Information on land
use and topography of these sites is provided in Table 5.

*Table 5 – Properties of the in situ sites used for the evaluation*

| Network (number of stations) | Location | Vegetation/climate | Soil texture | Topography |
|---|---|---|---|---|
| **Walnut Gulch Watershed** | Southeastern Arizona, USA | Brush- and grass-covered-Desert shrubs rangeland-Cattle grazing/ Semiarid | Range/sandy loam | Rolling |
| **Little Washita watershed** | Southwest Oklahoma, USA | Rangeland and pasture (63%), winter wheat / Sub humid | Range-wheat/silt or sand | Rolling |
| **AMMA Catch network Niger** | Niger | South Sahelian climate with semi-arid vegetation and crops (millet, fallows and tiger bush). | sandy loam, 91 % sand and 9% clay | - |
| **AMMA Catch network Ouémé** | Benin | Soudanian climate with different types of rain systems and Guinean savanna vegetation. | 77% sand and 19 % clay | - |


**5. Methodology of evaluation**
**5.1 Global comparison of SMOS and SMAP TB**
In order to compare SMOS TB product to SMAP TB, the SMOS daily product was averaged following
the same interpolation procedure as the one suggested in the SMAP mission. The method consists in using
an inverse distance weighting for all the SMOS EASE 2.0 at 25 km grids point in the limits of the EASE
2.0 at 36 km grid of the SMAP product. The $TB_H$ and $TB_V$ from SMAP product are extracted and used as
is. The comparison is done over the pixels with a water fraction of less than 0.001 (i.e. 0.1%) since the
SMAP TBs are provided with subtracted open surface water.




**5.2 Global Soil moisture maps comparison**
Global comparison is done over the EASE-Grid 2.0 25 km used for the L3 MO SM product. The L3 MO
SM field is extracted directly from the product. The L2 SO SM product is interpolated to the EASE-Grid
2.0 25 km using a three stage interpolation strategy where the availability of the products inside the limits
of the grid node is considered:
• bilinear, if more than two soil moisture retrievals are available.
• linear, if two soil moisture retrievals are available.
• nearest point, if one soil moisture retrieval is available.
The L2 SO SM is also filtered at high latitude where several soil moisture retrievals are available. The
selection criterion is minimum distance from the swath centre, the same as for the L3 MO SM algorithm.
**5.2 Local evaluations**
No interpolation is used after the extraction of the SM time series. The comparison is based on the
following statistical indicators:
- Mean bias ($m^3/m^3$)
- Standard Error of the Estimate (SEE) ($m^3/m^3$)
- Pearson correlation coefficient (R)
- Root mean square Error (RMSE) ($m^3/m^3$)

$$RMSE = \sqrt{\frac{1}{N}\sum_{i=1}^{N}(SM_{MO,i} - SM_{SO,i})^2} \qquad (4)$$

Where $SM_{MO,i}$ is the SM from multi-orbit retrievals and $SM_{SO,i}$ is the SM from single-orbit
retrievals.
- The empirical cumulative distribution function (Cox & Oakes, 1984).
**6. Results & Discussions**
**6.1 SMOS and SMAP Brightness temperatures**
Figure 5 (a,b) and Figure 6 (a,b) show the comparison between the SMOS L3 TB and SMAP L3 TB at 40°
incidence angle. Figure 5 (a) shows the average of SMOS and SMAP $TB_H$ and $TB_V$ for winter (Jan., Feb.,
Mar.) and summer (Jul., Aug., Sept.) seasons for year 2016. The gaps (in dark blue) in the SMOS images
are due to RFI with a differentiated impact for ascending and descending orbits. The difference in TBs
between H/V acquisitions is smaller than between ascending/descending configurations. The SMAP

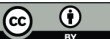



products show a higher coverage because SMAP has on-board RFI filtering and mitigation which enables
a better coverage but at the cost of a lower radiometric accuracy. The spatial patterns of TB are highly
consistent for the two missions. Figure 6 (a,b) show the distribution of difference of $TB_H$ and $TB_V$ from
SMOS and SMAP for winter (Jan., Feb., Mar.) and summer (Jul., Aug., Sept.) seasons during year 2016.
As described in Section 5.1, only nodes with a water fraction of less than 0.01 (i.e. 1 %) are considered.
The mean difference is about -3.67 K to -4.16 K with SMAP being colder independently of polarization or
season. The standard deviation of all comparisons is about 3.65 K. This value is due to differences in
calibration of the sensors and to the impact of differences in the acquisition time.

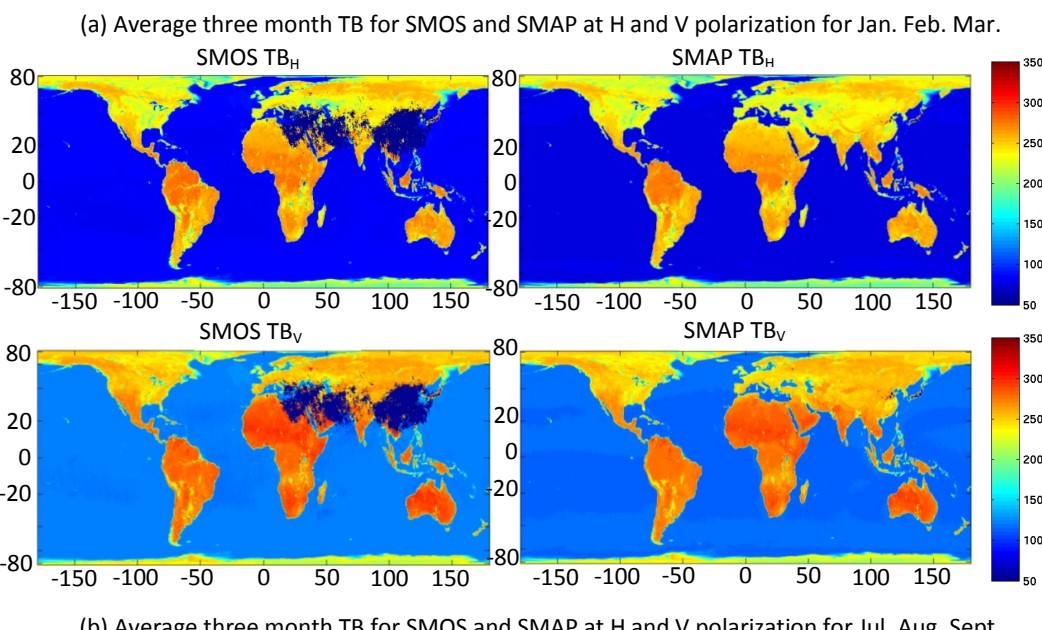

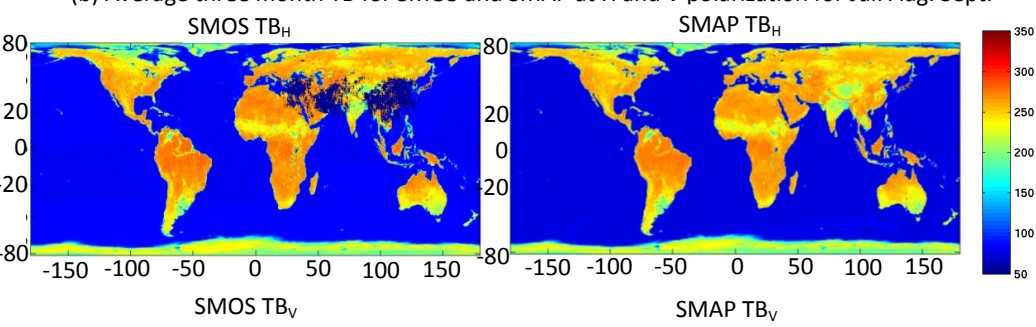

(a) Average three month TB for SMOS and SMAP at H and V polarization for Jan. Feb. Mar.

(b) Average three month TB for SMOS and SMAP at H and V polarization for Jul. Aug. Sept.



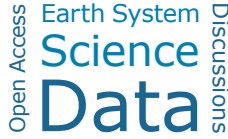

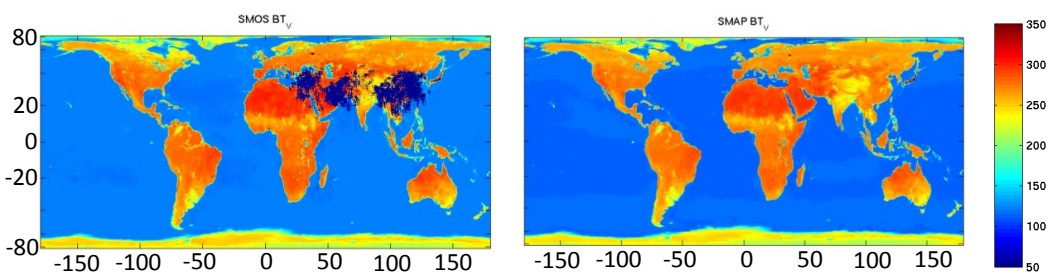

*Figure 5 – Three month average maps of SMOS L3 TB @40° (left) and SMAP L3 TB (right) for H*
*polarisation, V polarization considering winter: Jan., Feb., Mar. (a) and summer: Jul., Aug., Sept. (b)*
*seasons.*

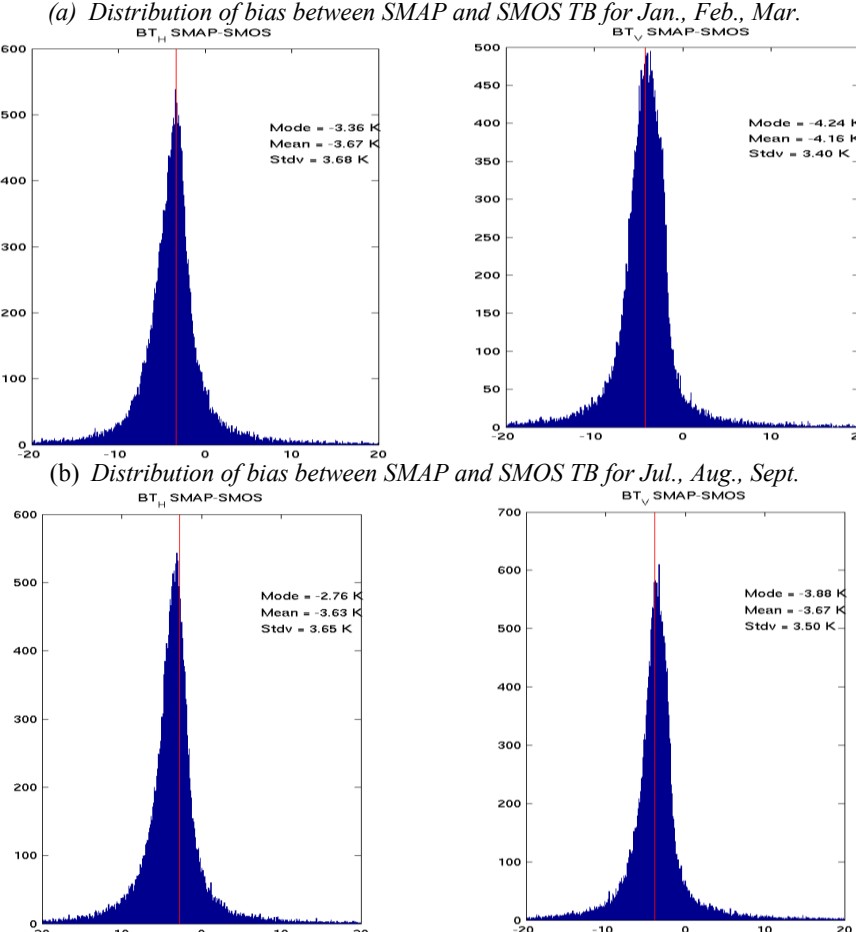

*Figure 6 – Distribution of bias between SMAP and SMOS L3 TB for pixels with less than 1 % of water*

*fraction for Jan. Feb. Mar. (a) and Jul. Aug. Sept. (b), H polarisation (right panel) and V polarisation*

*(left panel).*

### 6.2 Soil moisture retrievals at global scale

Based on the aforementioned evaluation methodology the L3SM MO retrievals are compared to those of

L2SM SO at global scale over the 2010-2015 period. The auxiliary maps of mean forest cover percentage

(Figure 8 a) and average RFI probabilities (Figure 8 b) for year 2011 are provided as complementary

information. These maps are obtained from the L3SM product. The mean forest cover (Figure 8 a)

provides the percentage of forest cover taking into account the mean antenna pattern. It is obtained by

convoluting the ECOCLIMAP (Masson et al., 2003) forest cover by the SMOS antenna weighting

function at a resolution of 4 km over an area of 125 × 125 km². The RFI map was obtained by averaging




the RFI probability field in the L3SM product. This information includes strong RFI and moderate RFI
depicted from the SMOS full polarization brightness temperatures (Richaume et al., 2014). Some soft and
mild RFI are not detected in this product.

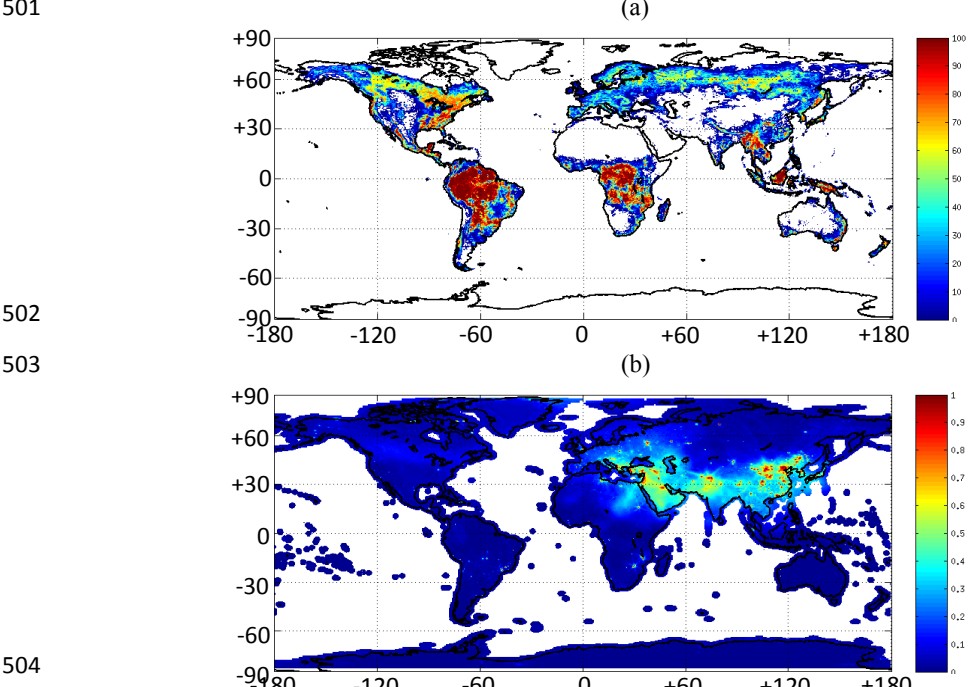

*Figure 7 – Global map of the mean forest cover percentage used in the SMOS L2S0 and L3MO soil moisture*
*retrievals (a) and map of the Radio Frequency interference (RFI) probabilities (b) for ascending orbit from the*
*L3MO soil moisture processor.*
Figures 8 (a,b) show the mean number of successful retrievals par year (2010-2015) obtained from L3SM
and L2SM respectively. White (Blank) pixels in Figure 8 (a) show the areas where no successful soil
moisture retrieval is available. These pixels are mostly located in areas of dense vegetation (Congo), area
that are seasonally inundated (Amazon Basin) and/or of high RFI (South-East Asia, Middle-East). From
Figures 10 (a) it is clear that the coverage area of the L3SM product is higher in these areas.





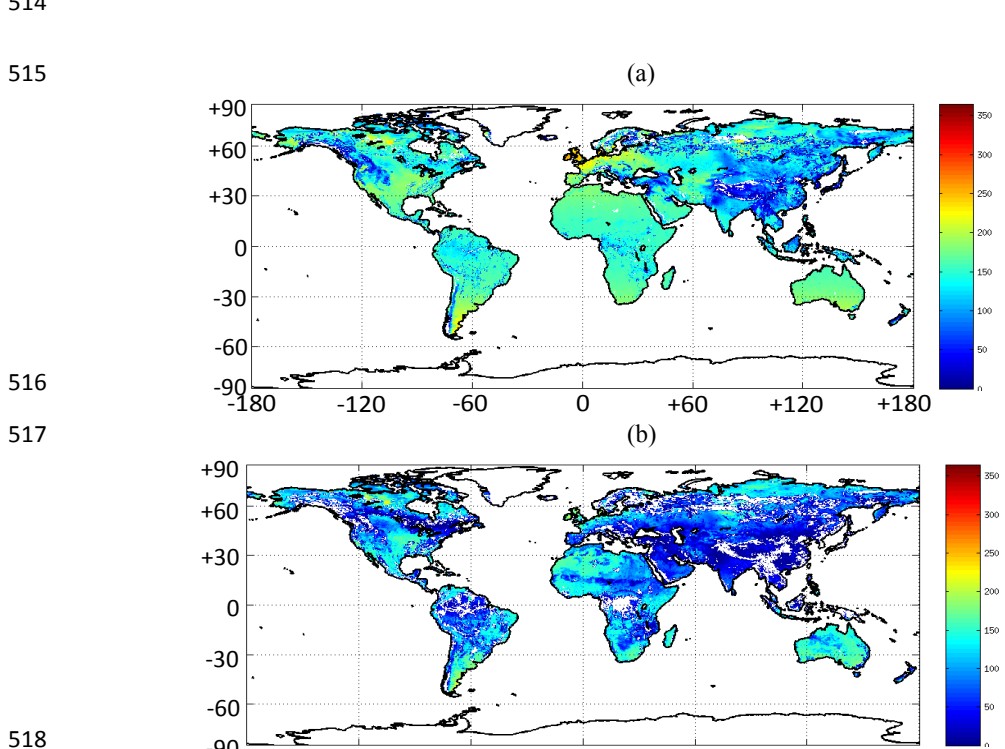




*Figure 8 – Mean number of successful SM retrievals per year  (2010-2015) for ascending orbits from L3SM-MO (a),*
*and L2SM- SO (b).*
Figures 9 (a,b) shows the difference (MO-SO) in the number of successful soil moisture retrievals
between L3SM and L2SM products. The general behaviour shows a systematic increase in the number of
retrievals. The number of retrievals is moderately increasing in desert and plain areas (10-20 retrievals /
year / orbits). The increase is much higher for forested areas. The L2SM showed a higher number of
successful retrievals in the area between 62°-70° longitude and 35°-55° latitudes. This is due to an
anomaly in the processing of TB products. The ancillary data containing the Total Electronic Content
(TEC) is not properly used over this region. This has been corrected and all operational products are now
properly processed. The archive products will be corrected for this error in the next processing campaign.
Also from Figure 13 it is clear that no enhancement in number of retrievals has been observed in areas
with very high RFI probabilities in descending orbits (not shown here) like the north Asia region.





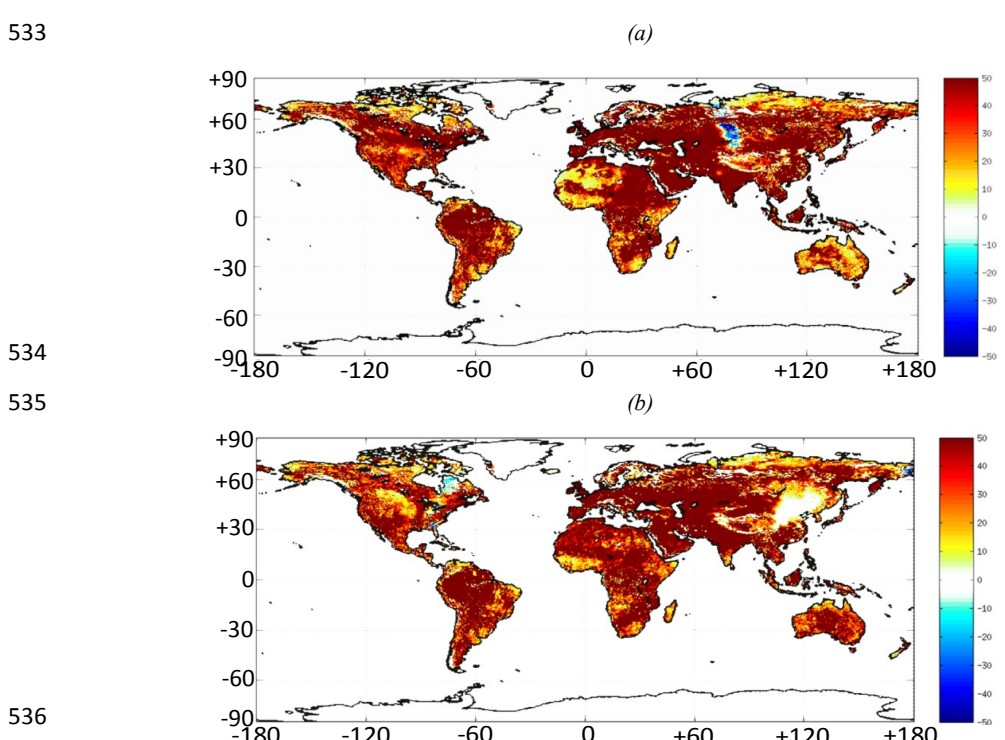

*Figure 9 – Global map of the difference in the mean number of SM successful retrieval per year over 2011-2015*

*(L3SM_{MO} - L2SM_{SO}) for ascending orbits (a) and descending orbits (b).*

The mean soil moisture from L3SM and L2SM for ascending orbits is provided in Figures 10 (a,b). The
figures show that the soil moisture spatial patterns are very similar between the SO and MO SM retrievals.
The coverage of the multi-orbit product is higher as already shown in the previous figures. Nevertheless
some discrepancies can be observed from the difference map (Figure 10 - c). The L3SM MO soil moisture
values are generally higher than those of L2SM SO. This is most visible in forested areas (Figure 7 - a)
which is consistent with climatic conditions over these areas. It is also higher in areas with high RFI
pollution (Figure 7-b). This leads in general to a decrease in the value of the retrieved soil moisture values.
So the higher L3SM can be due to the positive impact of using multiple dates during RFI prone periods.





(a)



(b)



(c)


*Figure 10 – Mean soil moisture map over 2011-2015 for ascending orbits from CATDS L3SM MO (a), DPGS L2SM*
*SO (b) and the difference (MO-SO) map between L3SM$_{MO}$ and L2SM$_{SO}$ (c).*
**6.2 *In situ* comparison**
The statistics for the comparison of L2SM SO and L3SM MO with *in situ* networks is shown in Table 3
and Table 4 for ascending and descending orbits respectively. The number of retrievals is systematically
better for the L3SM than L2SM as expected from the global analysis. Note that, contrary to the global



analysis, the *in situ* analysis is done without any grid interpolation by considering the closest node. The
skills are of similar magnitude for the LW and Niger sites and the lowest skill is obtained for the Benin
site in descending overpasses. No site showed lower number of successful retrievals for L3SM than The
bias values are not much improved by the L3SM. On the contrary they seem to increase in the majority of
the sites. The correlation values range from 0.65 to 0.88 for the different sites. Increased correlation was
found for the L3SM products over the Niger site and slightly over WG in descending overpasses. The
majority of the correlation values remain high with L3SM retrieval with no significant difference between
L2SM and L3SM.
Table 6 – Statistics of the in situ vs SMOS L3SM and L2SM for ascending orbits

| Site | R | | Bias ($m^3/m^3$) | | SEE ($m^3/m^3$) | | RMSE ($m^3/m^3$) | | Nb pt | |
|---|---|---|---|---|---|---|---|---|---|---|
| | L2 | L3 | L2 | L3 | L2 | L3 | L2 | L3 | L2 | L3 |
| **AMMA CATCH** **Benin** | 0.84 | 0.74 | -0.039 | -0.058 | 0.056 | 0.082 | 0.068 | 0.101 | 484 | 552 |
| **Niger** | 0.82 | 0.81 | -0.006 | -0.003 | 0.052 | 0.047 | 0.052 | 0.047 | 617 | 644 |
| **WATERSHEDS** **Little Washita** | 0.83 | 0.82 | -0.021 | -0.03 | 0.041 | 0.045 | 0.046 | 0.054 | 625 | 636 |
| **Walnut Gulch** | 0.81 | 0.73 | 0.005 | -0.007 | 0.038 | 0.053 | 0.039 | 0.053 | 638 | 643 |

*Table 7 – Statistics of the in situ vs SMOS L3SM and L2SM for descending orbits*

| Site | R | | Bias ($m^3/m^3$) | | SEE ($m^3/m^3$) | | RMSE ($m^3/m^3$) | | Nb pt | |
|---|---|---|---|---|---|---|---|---|---|---|
| | L2 | L3 | L2 | L3 | L2 | L3 | L2 | L3 | L2 | L3 |
| **AMMA CATCH** **Benin** | 0.74 | 0.61 | -0.029 | -0.037 | 0.069 | 0.104 | 0.075 | 0.11 | 636 | 667 |
| **Niger** | 0.63 | 0.65 | -0.011 | -0.008 | 0.049 | 0.049 | 0.05 | 0.05 | 540 | 598 |
| **WATERSHEDS** **Little Washita** | 0.81 | 0.80 | -0.001 | -0.012 | 0.042 | 0.043 | 0.042 | 0.044 | 333 | 364 |
| **Walnut Gulch** | 0.69 | 0.72 | -0.019 | -0.029 | 0.047 | 0.048 | 0.051 | 0.056 | 327 | 360 |

More in-depth analysis can be obtained by inspecting the times series of soil moisture. Figures 11 and 12
show the time series for the selected sites for the period 2010 to 2016 and for ascending and descending
overpasses. The Niger and Benin sites present a very pronounced seasonal signal typical of the Sahelian



sites. Over these sites the L3SM shows consistently lower soil moisture than L2SM for high soil moisture
values. The L3SM is closer in this case to the site data.  The time series for LW show that the SMOS data
closely follows the behaviour of the soil moisture dynamics over this site.  One of the reasons is that the
rainfall events are well separated enabling the remote sensing data to capture the dynamics of physical
processes like infiltration and evaporation at coarse scale. Thus the exponential behaviour typical of a
drying soil is well depicted.



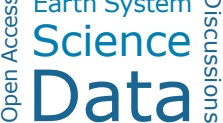

*Figure 11 – Time series for the validation sites for ascending overpasses.*


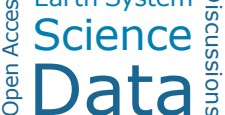

*Figure 12 - Time series for the validation sites for descending overpasses.*





Figure 13 and 14 show the CDF of the *in situ*, L2SM and L3SM data for ascending and descending orbits.
From these figures it can be concluded that the SMOS soil moisture is drier than the 5 cm *in situ* data
across the different values of soil moisture, this can be explained by the SMOS penetration depth with
respect to that of ground sensors. Nevertheless the shape of the distribution function, describing the
extreme and seasonal cycles, is well captured in most of cases. The Niger site Sahelian climate is well
captured with a high probability of low soil moisture values and low number of extreme values.  The
differences between the L2SM and the L3SM data are mainly observed for the Benin and LW sites. When
comparing figure 13 and figure 14 low differences can be notes between ascending and descending orbits.

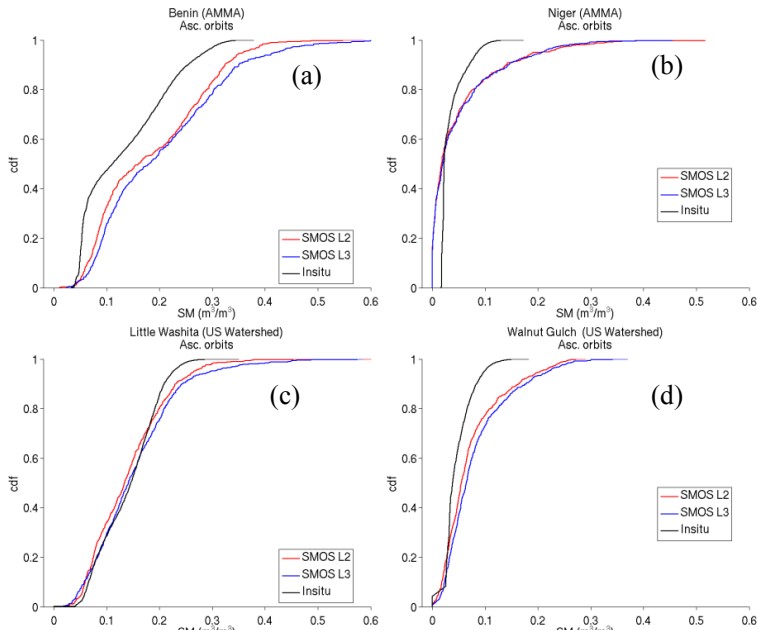

*Figure 13 – Cumulative Distribution Function (CDF) for the validation sites for ascending overpasses.*





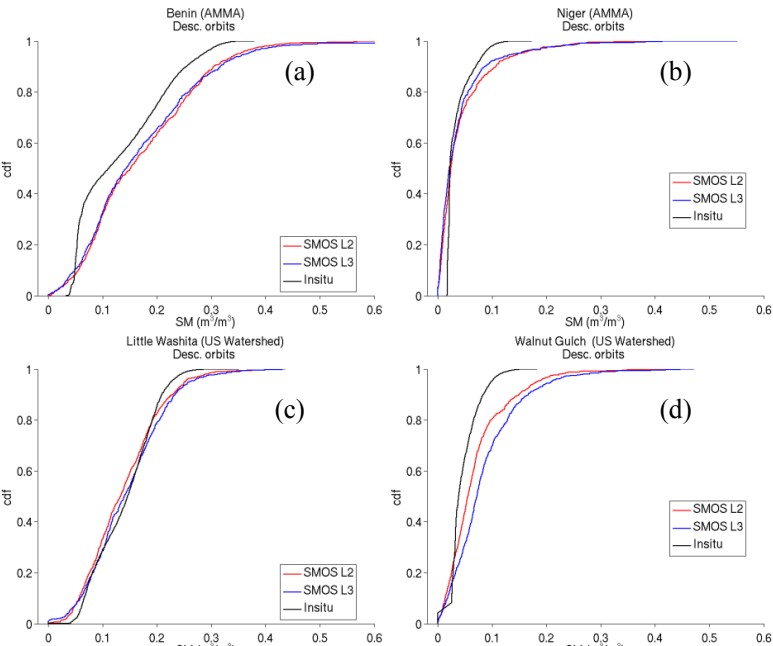

*Figure 14 - Cumulative Distribution Function (CDF) for the validation sites for descending overpasses.*

**7. Conclusions**
The level 3 daily maps of soil moisture and brightness temperatures are presented in this paper. A mutli-
orbit soil moisture retrieval algorithm for SMOS data is used to obtain the soil moisture product. The main
feature of the algorithm is the use of multiple revisits and of auto-correlation of optical vegetation depth in
the cost function. The algorithm is implemented operationally at CATDS. The processing chain delivers
gridded products over the EASE 2.0 grid at 25 km in NetCDF format. The L3 angle binned TB product is
compared to SMAP brightness temperature maps at 40°. The results show small differences in mean TB
between the products for H/V polarization and ascending and descending orbits. The SMAP product
presents a wider coverage due to the on-board RFI filtering. The L3 SM product is compared to the L2
SM product. The best improvements in algorithm performances are in terms of the number of successful
retrievals observed over forested and RFI prone areas. Also the L3SM product shows on average wetter
soil moisture retrievals than L2SM. The comparison with local sites showed that the quality of the
retrieval is comparable between L2SM and L3SM. This shows that the increase in the number of
successful retrieval does not degrade quality, but rather comes at the expense of an increased time lag in



product availability (6 hours for L2SM SO versus 3.5 to 7 days for L3SM MO ). Future works will
concentrate on the associated optical thickness product not presented in this paper. An application of the
algorithm to the SMAP data has been envisioned.

**Acknowledgements**
The SMOS L3SM products were obtained from the Centre Aval de Traitement des Données SMOS
(CATDS), operated for the "Centre National d'Etudes Spatiales" (CNES, France) by IFREMER (Brest,
France). This study was supported by the CNES "Terre, Océan, Surfaces Continentales, Atmosphère"
program. The authors would like to thank the USDA-ARS Hydrology and Remote Sensing Laboratory,
AMMA-Catch project for the *in situ* datasets.

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
