# Peer review of "The Global SMOS Level 3 daily soil moisture and brightness"

_Earth System Science Data, 2017_

## Referee Comment (RC1) · Anonymous Referee #1 · 15 Mar 2017

General comments This paper presents the SMOS L3 SM products. The paper is well structured and presents key elements for this dataset which is already used by numerous scientific groups. In this respect it was a needed paper. When comparing the L3 and L2 products with in-situ data, the statistics (RMSE, bias etc) for the L3 product are not improving the L2. The wet bias, from my point of view, is not sufficiently explained. I would suggest to deeper analyse the reasons for that bias and also to include in the conclusions some outlook to try to improve this.

Specific comments

L73 not clear what decadal time series means. 10 days periods, 10 years?

L77 Please precise that you refer to 'passive' microwave sensors (not active)

[Figure]

L170 - 178. Not clear what you are trying to explain here. It might seem the decrease of TB measurements is related to the calibration phase. Might be clear if you just plot the average number of measurements as a function of the swath position

L474- L475 Is it really higher difference between ASC/DESC passes than between polarizations. How can that be? And why? What about emissivities?

L508 Refers to ASC right? What about DESC?

L524 - 525 Only for ASC. Why?

---

## Referee Comment (RC2) · M. Schwank (Referee) · 20 Mar 2017

My general impression of the work is that is it definitively worth publishing as it contains valuable information on the quality and comparability of SMOS and SMAP data products of the same kind (soil surface moisture). However, I believe that the manuscript needs to be improved at many places to meet the standard of a scientific article, and also to accomplish didactical and linguistic issues. I do not expect the authors to implement all of the corrections and additions I have proposed as sticky-notes in the uploaded pdf "essd-2017-1__Rev". Nevertheless, I hope the comments are helpful for improving the overall quality of the manuscript on its way to the final publication.

Please excuse my late response to this review request. With kind regards.

[Figure]

Please also note the supplement to this comment:
http://www.earth-syst-sci-data-discuss.net/essd-2017-1/essd-2017-1-RC2-supplement.pdf

---

## Author Comment (AC1) · 8 Apr 2017

We thank the referee for his constructive comments. We provided the answers to his comments hereafter. We attached a rivised version of the manuscript that accounts for the comments of the two referees.

Comment : General comments This paper presents the SMOS L3 SM products. The paper is well structured and presents key elements for this dataset which is already used by numerous scientific groups. In this respect it was a needed paper.

Answer : We thank the referee for this positive comment. Indeed this dataset is widely used by the community either for the soil moisture or the brightness temperature products.

Comment: When comparing the L3 and L2 products with in-situ data, the statistics (RMSE, bias etc) for the L3 product are not improving the L2. The wet bias, from my point of view, is not sufficiently explained. I would suggest to deeper analyse the reasons for that bias and also to include in the conclusions some outlook to /try to improve this.

Answer: The statistics in table 5 and 6 show that the RMSE, R and SEE of L3 and L2 datasets are quite similar but they also show that main improvement of L3 was the increase in the number of retrievals as mentioned in the manuscript. There are very few studies that have used both the L2 and the L3 SMOS data to evaluate them with respect to in situ measurements. However, the results presented here are in perfect agreement with those of Kerr et al. 2016 (RSE).

On the issue of bias: Considering that the bias is computed as (in-situ – retrieved soil moisture) the negative bias corresponds to drier soil moisture retrievals. This was clarified in the text. The absolute values of bias are smaller than 0.04 m3/m3 except for the Benin site in ascending orbits. Considering the local measurement errors, the difference in the representative depth (sensing depth) and the impact of spatial heterogeneity, the bias values can be considered as low. For instance Vaz et al. 2013 estimates the sole errors from in-situ sensor of about 0.015 to 0.025 m3/m3 . Nevertheless we agree with the referee that the bias is in one direction and more investigation is needed unfortunately there is not a single simple answer to that. SMOS SM can show a wet or dry bias with respect to in-situ measurements depending on the sites. See for instance the barplots of Figure 5c of Kerr et al. (2016). Notice that on average the bias is towards drier values than the in-situ measurements but that there are positive and negative values. The same behaviour has been found in other studies see for instance Al Bitar et al. (2012) and Table 2 in Al-Yaari et al. (2017 RSE), where the mean bias for different in situ measurements networks is positive of some networks and drier for others (both for SMOS L3 and applying LPRM to SMOS brightness temperatures). Very detailed studies at site level would be needed to understand if the bias is just a con-

sequence of these representativity effects or in can be reduced for instance using very high spatial resolution land cover auxiliary data. For instance Draper et al.( 2012) listed the reasons of the bias between in-situ and remote sensing based soil moisture from AMSR-E. Unfortunately that kind of studies is beyond the scope of this paper, which is devoted to the presentation and discussion of a global dataset. Following the referee comments, the manuscript has been modified accordingly including the complementary information given above.

References:

A. Al-Yaari, J.-P. Wigneron, Y. Kerr, N. Rodriguez-Fernandez, P.E. O'Neill, T.J. Jackson, G.J.M. De Lannoy, A. Al Bitar, A. Mialon, P. Richaume, J.P. Walker, A. Mahmoodi, S. Yueh, Evaluating soil moisture retrievals from ESA's SMOS and NASA's SMAP brightness temperature datasets, Remote Sensing of Environment, Volume 193, May 2017, Pages 257-273, ISSN 0034-4257, http://dx.doi.org/10.1016/j.rse.2017.03.010.

Clara S. Draper, Jeffrey P. Walker, Peter J. Steinle, Richard A.M. de Jeu, Thomas R.H. Holmes, An evaluation of AMSR–E derived soil moisture over Australia, Remote Sensing of Environment, Volume 113, Issue 4, 15 April 2009, Pages 703-710, ISSN 0034-4257,

Kerr, Y.H., Al-Yaari, A., Rodriguez-Fernandez, N., Parrens, M., Molero, B., Leroux, D., Bircher, S., Mahmoodi, A., Mialon, A., Richaume, P., Delwart, S., Al Bitar, A., Pellarin, T., Bindlish, R., Jackson, T.J., Rüdiger, C., Waldteufel, P., Mecklenburg, S., Wigneron, J.-P. Overview of SMOS performance in terms of global soil moisture monitoring after six years in operation, Remote Sensing of Environment, Volume 180, July 2016, Pages 40-63, ISSN 0034-4257, http://doi.org/10.1016/j.rse.2016.02.042.

Vaz, Carlos M.P. and Jones, Scott and Meding, Mercer and Tuller, Markus, 2013. Evaluation of Standard Calibration Functions for Eight Electromagnetic Soil Moisture Sensors,Vadose Zone Journal, vol. 12 , nb 2, DOI 10.2136/vzj2012.0160.

Specific comments Comment: L73 not clear what decadal time series means. 10 days periods, 10 years?

Answer: The ESA CCI aims at generating long time series in support of climate change studies and here decadal means 10 years. We clarified in the text.

Comment: L77 Please precise that you refer to 'passive' microwave sensors (not active)

Answer: Done

Comment: L170 - 178. Not clear what you are trying to explain here. It might seem the decrease of TB measurements is related to the calibration phase. Might be clear if you just plot the average number of measurements as a function of the swath position

Answer: Two reasons of change in TB numbers are depicted in the panels. The first is due to the calibration phase leading to a decrease of number of TBs. This is of low impact. The second is due to the acquisition configuration or the reduced number of TBs when the point of interest is at the limit (or border) of the swath. This can lead to failure in the retrievals. We updated the text for clarity.

Comment: L474- L475 Is it really higher difference between ASC/DESC passes than between polarizations. How can that be? And why? What about emissivities?

Answer: Yes, the differences in TB for ascending and descending orbits are higher compared to the differences of TB at different polarizations. This can be associated to two reasons: First, the L1 algorithm in SMAP and SMOS does not use the same configuration for the computation of the Faraday rotation. The Faraday rotation is impacted by the TEC (Total Electronic Content) in the ionosphere. SMAP algorithm uses the Stokes 3 parameter to account for the Faraday rotation, while auxiliary TEC files are used in the SMOS algorithm to compute the Faraday rotation. The ionosphere TEC is very different between ascending and descending orbits as the heating during the day results in an increase of the TEC in the afternoon. The second is that the RFI probabilities are very different between ascending and descending orbits due to directional aspects and they are closer between H/V polarizations. This explanation was added to the text.

Comment: L508 Refers to ASC right? What about DESC?

Answer: We present here the results from descending orbits from SMAP that coincide with the 06H00 AM local timing against the ascending orbits from SMOS which are also at 6H00 AM. The SMAP mission is not delivering L3 product for ascending orbits which coincide with 18h00 local timing. The Ascending orbits are only delivered at L1. L1 are not time synthesis (global maps) at Top of atmosphere and not readily comparable with L3 SMOS data. We clarified this in the text.

Comment: L524 - 525 Only for ASC. Why? Answer: Please check answer above.

Please also note the supplement to this comment:
http://www.earth-syst-sci-data-discuss.net/essd-2017-1/essd-2017-1-AC1-supplement.zip

---

## Author Comment (AC2) · 8 Apr 2017

We thank Dr. Schwank for his detailed analysis of the paper and for providing constructive comments to improve the paper. We hope the modification met approval.

Comment: My general impression of the work is that is it definitively worth publishing as it contains valuable information on the quality and comparability of SMOS and SMAP data products of the same kind (soil surface moisture). However, I believe that the manuscript needs to be improved at many places to meet the standard of a scientific article, and also to accomplish didactical and linguistic issues. I do not expect the authors to implement all of the corrections and additions I have proposed as sticky-notes in the uploaded pdf "essd-2017-1__Rev". Nevertheless, I hope the comments are helpful for improving the overall quality of the manuscript on its way to the final

publication. Please excuse my late response to this review request. With kind regards.

Answer: We took into consideration all the modifications suggested in the pdf file and modified the word document accordingly. We also provided the answers of the open questions as comments in the pdf file.

Please also note the supplement to this comment:
http://www.earth-syst-sci-data-discuss.net/essd-2017-1/essd-2017-1-AC2-supplement.zip